# An Exploratory Study on the Diverse Uses and Benefits of Locally-Sourced Fruit Species in Three Villages of Mpumalanga Province, South Africa

**DOI:** 10.3390/foods9111581

**Published:** 2020-10-31

**Authors:** Kutullo Nick Shai, Khayelihle Ncama, Peter Tshepiso Ndhlovu, Madeleen Struwig, Adeyemi Oladapo Aremu

**Affiliations:** 1Indigenous Knowledge Systems Centre, Faculty of Natural and Agricultural Sciences, North-West University, Private Bag X2046, Mmabatho 2745, North West Province, South Africa; Kutullo.Shai@nwu.ac.za (K.N.S.); Tshepiso.Ndhlovu@nwu.ac.za (P.T.N.); 2Department of Crop Sciences, Faculty of Natural and Agricultural Sciences, North-West University, Private Bag X2046, Mmabatho 2745, North West Province, South Africa; 31390927@nwu.ac.za; 3Unit for Environmental Sciences and Management, Faculty of Natural and Agricultural Sciences, North-West University, Private Bag X6001, Potchefstroom 2520, South Africa; madeleen.struwig@nwu.ac.za; 4Food Security and Safety Niche Area, Faculty of Natural and Agricultural Sciences, North-West University, Private Bag X2046, Mmabatho 2745, North West Province, South Africa

**Keywords:** beverages, bio-economy, conservation, food security, indigenous knowledge, wild edible plants

## Abstract

Globally, the potential of indigenous and neglected fruit species is continuously being recognized. In the current study, we explored the uses and benefits of locally available fruit species among the Mapulana people in Bushbuckridge Local Municipality. An ethno-botanical survey was conducted using in-depth interviews to record the names of the fruit species, their uses, seasonal availability, and occurrence in three villages, namely, Mokhololine, Motlamogatsane, and Rooiboklaagte B. Forty-one (41) participants aged 23 to 89 years old, identified by community members as knowledgeable on the utilization of fruit species, were interviewed. The frequency of citation (FC), use value (UV), and use report (UR) of the locally sourced fruit species were determined. The study revealed thirty-one (31) indigenous/naturalized plants belonging to 17 families with Anacardiaceae (four species) and Rubiaceae (three species) as the dominant ones. Approximately 48% of the 31 plants had FC of 100%, suggesting their high popularity in the study area. The identified plants had diverse uses that were categorized into six (6) groups and mainly dominated by food (59%) and medicine (34%). *Strychnos madagascariensis* had the highest (0.56) UV while *Berchemia discolor*, *Parinari capensis*, *Parinari curatellifolia,* and *Sclerocarya birrea* had the highest (6) URs. Overall, these locally sourced fruit species still play a significant role in the daily lives of the Mapulana people. The identified fruit species have the potential to be considered as alternative sources to meet the dietary requirements and health needs, especially in rural communities.

## 1. Introduction

Globally, food and nutritional insecurity remain among the biggest concerns facing humans, as recently highlighted by Willett et al. [1]. Likewise, food deficiency and insecurity remain a major challenge in many rural communities, especially in developing countries [2,3,4]. Since time immemorial, local communities have relied on diverse native plant species because they are easily accessed from their immediate environment. It is undeniable that the indigenous fruits contribute to food security, especially in developing countries [5,6]. Particularly, sub-Saharan Africa is endowed with numerous indigenous fruit species that can be explored and incorporated into the diets to solve nutrition-related concerns [4,7,8]. 

Local communities continue to utilize and consume indigenous edible fruits for food and medicine as well as fulfilling the socio-cultural needs and livelihood of the people [6,9,10,11]. Particularly, it may be a supplementary income or the major income for the household [12,13]. As part of the benefit associated with indigenous fruit trees, they are well adapted to local environmental conditions and often withstand abiotic and biotic stresses better than introduced plants [6,10]. Furthermore, Awodoyin et al. [14] stated that encouraging the domestication and protection of indigenous fruit trees in the agro-ecosystem may provide highly sustainable systems that maintain soil productivity, micronutrients, and biodiversity. This can enhance food productivity, adaptation, and mitigation to climate change. As a result, Cemansky [13] articulated the need for concerted research effort and policies to mitigate the ongoing decline of indigenous fruit trees.

Exploring the potential of indigenous fruit trees for sustainable livelihoods of humans remains a research priority in several parts of the world [1,4,11,15]. In South Africa, many local communities in the different provinces, especially Mpumalanga, acknowledge the value of plant biodiversity as an integral part of their culture for survival and general welfare [16,17]. Similarly, there is a growing awareness of the importance of indigenous plants, especially fruits, for food security and health-promoting value [4,8,18]. Given the high occurrence and diversity of indigenous (native or wild) fruits in many South African communities, they remain an affordable and potential source of meeting the daily nutritional requirement especially for the essential vitamins and minerals [18,19]. However, the indigenous knowledge on these plants is diminishing due to urban migration as well as the high preference for exotic fruits. This has created a knowledge transmission gap between the old and youthful generations [20,21]. Exploring indigenous knowledge on the uses of native plants provide the basis to pursue their preservation and promotion locally, nationally, and internationally [8,22,23]. Moreover, these plants and their associated products have the potential to contribute toward achieving the Malabo Declaration goals 2025 and United Nations Sustainable Development Goals (SDG) Agenda 2030 such as zero hunger and no poverty, as well as good health and well-being. Thus, the current study explored the indigenous knowledge on the diverse uses and benefits of locally sourced fruits (wild/indigenous and naturalized) among the Mapulana people in Bushbuckridge Local Municipality. The following research questions guided the study:Which locally sourced fruit trees are found among the Mapulana people in Bushbuckridge Local Municipality?How do Mapulana people use and benefit from the wild fruit trees in their locality?

## 2. Materials and Methods 

### 2.1. Study Area

The study was conducted in Setlhare Tribal Council in Bushbuckridge Local Municipality, Ehlanzeni District of South Africa. The Ehlanzeni District is situated in the northeast of Mpumalanga Province, demarcated by Mozambique and Swaziland, and covers a surface of 27,895.47 km^2^. Rural areas make up the most of Bushbuckridge region, with small rural villages containing 29% of the populace and dense rural villages represent 61% [24]. It has 34 wards and more than 135 villages, with an estimated of 509,964 people, who are predominantly (99.5%) black Africans, based on data from Stats SA [24]. The Mapulana people found in the three selected villages are descendants of the Northern Sotho ethnic group of Lebowa [25]. 

As highlighted by Shackleton et al. [12], Bushbuckridge has the mean annual rainfall of approximately 1200 mm in the west and diminishing to 550 mm in the east. The area has no frost and the mean annual temperature is approximately 22 °C [21]. The primary crops cultivated consist of groundnuts, maize, and various beans. A majority of households harvest various resources such as fruits, firewood, thatch grass, mushrooms, and reeds from their immediate surroundings [12]. The area is often negatively impacted by challenges such as dry spells, veld fires, and non-ecofriendly cultivating practices [21,24]. For the current study, three (3) villages, namely, Mokhololine (24°40′45.96″ S and 30°51′18.98″ E), Motlamogatsane (24°38′22.02″ S and 31°1′21.22″ E), and Rooiboklaagte B (24°39′42.12″ S and 31°3′32.04″ S) were sampled due to their long history of dependence on locally sourced fruits (Figure 1). 

### 2.2. Ethno-Botanical Survey

The ethno-botanical information on plants was collected from December 2018 to January 2019. The information was collected using in-depth interview guides where details such as the names, uses, and the availability of the fruit trees were recorded. The interview sessions were conducted in Sepulana (Northern Sotho sublanguage) with the assistance of a field assistant who is fluent in both English and Sepulana. A total of 41 participants were interviewed from the three (3) selected villages. The identification and recruitment of participants were done in consultation with the traditional leader of the selected communities. The plants mentioned by the participants were collected with the help of the research assistant during the field walks. Thereafter, voucher specimens were prepared and deposited at the S.D. Phalatse Herbarium, North-West University, Mahikeng Campus (UNWH) and the National Herbarium, Pretoria (PRE), South Africa. 

### 2.3. Data Analysis

To determine the importance of the locally sourced fruit trees in the study area, we applied two (2) ethno-botanical indices including frequency of citations (FC) and use value (UV), based on the description by Tardío and Pardo de Santayana [26]. 

#### 2.3.1. Frequency of Citation

The percentage of participants claiming the use of a particular fruit tree was calculated as follows:(1)$$FC=\left(\frac{Np}{N}\right)\times 100$$
where *Np* is the number of citations for a particular fruit tree, and *N* is the total number of participants in the study.

#### 2.3.2. Use Values (UV)

The use value is an ethno-botanical index that shows the relative importance of plant species known locally based on the number of recorded uses for each species. It was calculated by following the formula given by Logan [27].
(2)$$Use\;value\;\left(UV\right)=\sum \frac{Ui}{N}$$
where *UV* stands for use value and “*Ui*” refers to the total number of uses per species while “n” is the number of participants who reported on the plant species.

#### 2.3.3. Use Categories and Use Report (UR)

The uses for the different plants were categorized following the guideline by Cook [28]. All the uses as reported by the participants were classified into one of the 13 levels provided in the guideline.

### 2.4. Ethical Approval

The study was approved (NWU-00601-18-A9) by the Faculty of Natural and Agricultural Sciences (FNAS) Research Ethics Committee, North-West University (NWU). Permission to access the study areas was provided by the Setlhare Tribal Authority of Bushbuckridge Local Municipality and written consent was obtained from all participants prior to the interviews. The permit to collect plants was granted under the National Forest Act, Act No. 84 of 1998 by the Department of Agriculture, Fish and Forestry, South Africa. 

## 3. Results and Discussion 

### 3.1. Demographic Characteristics of the Participants 

A total of 41 participants with diverse demographic characteristics was involved in the current study (Table 1). Among the 41 individuals, 24% were self-employed in the study area. The participants knowledgeable on the utilization of plant species ranged from 23 to 89 years. The participants were categorized into three groups, namely, young (aged 18–35: n = 9), middle-aged (aged 36–64: n = 13), and elderly (aged > 64: n = 19). The majority (58.5%) of the participants were female. The youth, who contributed approximately 22% of the participants, had little enthusiasm for indigenous knowledge on the utilization of fruit species. As a result, there is a risk of knowledge loss if nothing is done to motivate and engage the youthful generation about the use of fruit species. Magwede et al. [20] emphasized the need for the preservation of indigenous knowledge that runs the risk of being lost due to the adoption of modern lifestyles. For instance, youthful individuals are influenced by globalization and, consequently, not keen on learning and participating in ethno-medicinal and cultural activities [29].

### 3.2. Inventory and Uses of Locally Sourced Fruits in the Study Area

A total of 31 plant species with multiple uses was identified in the current study (Table 2). The importance of fruit trees/shrubs among local communities has been documented in countries such as Namibia [11], Zimbabwe [30], Ethiopia [31], India [32,33], Estonia [34], and Indonesia [35]. These aforementioned studies suggest that local communities often rely on locally sourced plants for diverse uses and as a source of livelihood. As indicated in Table 2, 100% FC were recorded for 15 fruit species (e.g., *Annona senegalensis*, *Carissa edulis*, *Diospyros mespiliformis*, *Flueggea virosa*, *Sclerocarya birrea*, *Ximenia caffra*), which suggests their popularity among the participants. Interestingly, *Sclerocarya birrea* and *Diospyros mespiliformis* were identified among the four top plants with the highest priority (based on the high number of citations among participants) in Namibia by Cheikhyoussef and Embashu [11]. On the other hand, six of the fruits (*Berchemia discolor*, *Searsia pendulina*, *Berchemia zeyheri*, *Bridelia micrantha*, *Strychnos madagascariensis,* and *Ximenia americana*) had the lowest FC (<40%) in the current study. 

In a recent study, Mashile et al. [21] demonstrated the increasing importance and popularity of fruit species in 15 villages located in three (Bushbuckridge, Mbombela, and Thaba Chweu) local municipalities in Mpumalanga Province. Although the majority (about 84%) of the fruit species in the current study were recently reported [21], five species, namely, *Canthium inerme*, *Macrotyloma maranguense*, *Searsia pendulina*, *Syzygium intermedium,* and *Trichilia emetica* were indicated for the first time in the study area. It was also evidenced that the type of fruits in the study area is different than other parts of South Africa where similar ethno-botanical surveys for indigenous food/fruit have been conducted. For instance, 28 indigenous fruits were documented among the contemporary Khoe-San descendants on the south coast of the western Cape Province [36]. However, none of these fruits was mentioned in our study among the Mapulana of Mpumalanga Province. Welcome and van Wyk [7] demonstrated the diverse patterns in the utilization of indigenous plants for food type among the different ethnic groups in southern Africa. 

In the current study, the top three fruit species in terms of UV were *Strychnos madagascariensis* (0.56), *Bridelia micrantha* (0.51), and *Berchemia discolour* (0.4). These aforementioned fruit species had the most common uses among a sizable number of participants. On the other hand, *Macrotyloma maranguense*, *Mimusops zeyheri,* and *Syzygium intermedium* had the least UV of 0.02, suggesting limited knowledge on their utilization among the participants. Currently, these aforementioned species are only consumed as edible fruit in the study area.

Diverse uses of the fruit species were evident in the study area. For instance, the consumption of all the fruits was common as a source of nutrient and energy. Plants such as *Annonas senegalensis*, *Trichilia emetica*, *Parinari capensis,* and *Ximenia caffra* are popular species used for different health conditions in the study area. Branches from plants such as *Diospyros mespiliformis*, *Lannea edulis,* and *Lannea schweinfurthii* are used for the production of wooden spoons as a useful utensil among the households. The use of species such as *Sclerocarya birrea*, *Pirinari capensis*, *Strychnos madagascariensis,* and *Vangueria infausta* for the production of different beverages (wine and juice), porridge, jam, juice, and sweets was common among the participants. In recent times, an increased awareness of the importance of indigenous plants for new product development and numerous new products have been demonstrated [8].

Despite the diverse uses for the identified fruit species, only six (food, materials, fuel, social uses, vertebrate poison, and medicine) out of the 13 categories were covered based on the guideline by Cook [28]. The utilization of indigenous plants is often low due to inadequate knowledge of their potential and existing undesirable traits as well as the inherent challenges associated with their production [4,14,23]. Nevertheless, many fruit species play an important role by contributing to the welfare and livelihoods of households in many South African rural communities [21,37]. Many wild, edible plants are important sources of vitamins and other nutrients for the indigenous communities [8,14,38].

### 3.3. Distribution of Plants Based on Families and Parts

In total, the identified plants were from 17 families with Anacardiaceae and Rubiaceae as the most dominant ones (Figure 2). Most of these 17 families are widespread and often used as wild foods in southern Africa [7]. The current results are similar to the study by Maroyi [30], whereby Anacardiaceae had the highest number of wild, edible plants in the Nhema communal area, Midlands Province, Zimbabwe. Likewise, Anacardiaceae and Rhamnaceae were the most dominant families for the 52 fruit plants documented across 15 villages located in Ehlanzeni district municipality of Mpumalanga Province, South Africa. On the other hand, Dejene et al. [31] reported families such as Malvaceae and Moraceae as common plant families of wild, edible fruit tree species in lowland areas of Ethiopia. Furthermore, the families occurring in the current study are similar to those reported by Mashile et al. [21]. Particularly, the Sapotaceae are considered as important fruit trees frequently used among communities in Mpumalanga Province, South Africa [21]. 

The current survey revealed that six (6) plant parts were used for diverse purposes (Figure 3). The four most frequently used parts were fruit (49%), roots (20%), bark (14%), and leaves (9%). This was similar to other studies [30,32], where plant parts such as fruit and leaves were among the most common plant parts. Based on the extensive analysis of the food plants of southern Africa by Welcome and van Wyk [7], fruits and leaves were identified as the most important plant parts used as food for most of the cultural groups. In the current study, the high incidence of root and bark is linked to the use of such plants for therapeutic purpose among the participants. Likewise, the roots were the most popular plant parts recorded for the cumulative for the indigenous fruit trees in Ohangwena and Oshikoto regions of Namibia [11].

### 3.4. Category of Uses for the Fruit Species among the Local Communities

Wild plants form an important part of the human diet with a significant number of plant species known to be edible [7]. In South Africa, the diverse uses of indigenous fruits among rural households have been extensively documented [5,16,20,21]. Social values held by the different communities have a significant effect on the manner in which wild, edible plants are consumed and conserved by the indigenous communities [39]. In the current study, the majority of these fruit species were considered as food (59%) and medicine (34%) while a few (7%) were utilized for other purposes such as fuel, material, vertebrate poison, and social uses (Figure 4). Generally, indigenous fruit species have been part of their diet whenever they are available, especially during their fruiting seasons [39]. Suwardi et al. [40] articulated that the locals tend to use certain parts of edible fruits to support some daily living requirements. Particularly, edible fruit trees may serve as an alternative food supply for households during food shortage [41]. As indicated by the participants, most of these fruit species have been part of their diet for a long time. The identified by-products derived from the locally sourced fruit species are prepared using diverse methods. For instance, the nuts found in the kernel of *Sclerocarya birrea* are sun-dried and ground into a powder to make a traditional snack called “*Lekoma*”. The fresh fruits are peeled off and eaten in a fresh state. Likewise, *Vangueria infausta* is sun-dried and eaten as snack bites. However, Mapulana does not use only the abovementioned species to make the “*Lekoma*” traditional snack; other plant species such as *Strychnos madagascariensis* are ground, sun-dried, and eaten in powder form [21]. 

As reported by Shackleton et al. [12], the wild fruits in the Bushbuckridge savanna region have diverse uses. For instance, *Sclerocarya birrea* is used for beverage, cleansers, and cooking oil. Fruits such as *Carissa edulis*, *Vangueria infausta*, *Vangueria pygmaea,* and *Sclerocarya birrea* are traditionally eaten as snacks and household-fermented beverages [5,20,21]. Among the 31 fruit species documented in the study area, the majority (>60%) are identified as popular and used as beverage-making species in villages of South Africa [8,20]. In South Africa, *Sclerocarya birrea* are gathered by rural communities for nourishment and making marula, a customary matured drink [21]. In the study area, it is known as “Borula” and is popular during the summer season. In Ethiopia, edible fruit trees such as *Carissa edulis* and *Ximenia americana* are often consumed as a supplementary diet [31]. 

Even though many indigenous/wild fruits are highly desired as food and dietary supplements [1,42], their therapeutic properties have been recognized. For instance, in northern Namibia, 76% of the 25 indigenous fruit trees had different uses in traditional medicine [11]. A significant proportion of the fruit species mentioned by the participants was utilized as remedies against 22 different ailments. Some of these fruit species possess medicinal purposes, as demonstrated in a previous study [43]. Plant species with the highest number of medicinal uses included *Bridelia micrantha, Strychnos madagascariensis,* and *Vangueria infausta*. The abovementioned plants were highly ranked and regarded as the most important for the treatment of various diseases such as sexually transmitted infections/diseases (e.g., syphilis and gonorrhea), snakebites, and excessively irritated bowel movements [17]. In this study, the knowledge holders mentioned that the role of wild fruits in treating diseases is crucial in the community. According to Tshikalange et al. [17] and Van Wyk and Gericke [44], numerous sicknesses and conditions such as tuberculosis, flu, toothache, snakebite, syphilis, gonorrhea, epilepsy, infertility, headache, and diarrhea are often treated with fruit species. In addition, the participants stated an example of treating gonorrhea and syphilis with a concoction of *Flueggea virosa, Sclerocarya birrea, Vangueria infausta,* and *Vangueria pygmaea.* The traditional health practitioner soaks and boils the plant parts in warm water for a particular period and demonstrated as an effective herbal remedy for the aforementioned condition.

The study also revealed the value of the fruit species in mitigating and boosting the health issues of community members. For instance, the majority of male participants highlighted the importance of frequently drinking a few glasses of brewed marula. For instance, the fruit can boost sex drive. As such, men are encouraged to drink the marula alcoholic beverage. Moreover, participants affirmed that they advise patients to feed on wild fruits so that they would obtain essential energy and nutrients. Likewise, Mogale et al. [45] highlighted the medicinal use of indigenous fruits in Central Sekhukhune, Limpopo province. 

### 3.5. Seasonal Availability and Occurrence of Fruits in the Study Area

Generally, most fruit species are accessible in the summer, i.e., hot-dry season, which is often associated with the scarcity of food supplies (Table 3). For some of the plants, the fruiting period indicated by the participants aligns with the existing evidence in the literature [5,46,47,48,49,50,51]. On the other hand, the extended fruiting duration reported for plants such as *Englerophytum magalismontanum*, *Ficus sur,* and *Ficus thonningii* was in contrast to shorter duration reported in literature (Table 3). Fruit harvesting seasons for fruits are known to slightly vary across different locations [31]. Studies have demonstrated that fruit harvesting from the wild and semi-domesticated trees that grow on farms can improve rural employment [10,11,12,13]. Particularly, the processing of indigenous fruits into jams and juices during the fruiting season may generate extra income [10]. In Africa, fruit species play an important role during the dry season and at the beginning of the rainy season in many local communities [12]. In the study area, the participants alluded to the availability of locally sourced fruits as species-specific (Table 3). As mentioned by one of the participants, “*We get them all seasons from summer, winter, autumn, and spring. Their occurrences may differ according to seasons. Like, the rainy season has its own blossom of wild fruits.*” For example, *Ficus sur* are available all year round while others occur in a specific season. An interesting finding of the current study was that some fruits are utilized as hunger quenchers and harvested during months of low agricultural productivity. The participants reported that some fruit species are commonly harvested during periods of food shortages, these included *Annona senegalensis, Carissa edulis, Diospyros mespiliformis, Ximenia caffra, Mimusops zeyheri,* and *Sclerocarya birrea*.

Maroyi [30] highlighted that fruiting seasons of wild, edible plants vary and the consumption alters with other species in terms of appearances. Based on the response from the participants, every season provides a constant supply of food resources for the community. As a result, many rural communities depend on wild, edible plants and related resources for their sustenance, especially in times of food shortage [4,12,21,30]. Winter collection coincides with the time of resting, festivities, migrations and displacements, working on new homes and redesign of old ones, visits to family members and companions, and numerous exercises that are just conceivable when individuals are briefly liberated from farming activities. Particularly, people in the indigenous communities have sufficient opportunity to walk and gather the indigenous fruits, given that they are free from agricultural activities [52].

### 3.6. Implication of Current Findings for Food and Nutritional Security

As a result of the increasing negative effect of unhealthy food on the health and well-being of humans as well as the severe impact on the environment, the importance and urgent need to collectively transform diets and food production cannot be overemphasized [1,4]. Interestingly, these challenges may be mitigated using the recently proposed reference diet that highly resonates with traditional eating pattern [1]. Furthermore, indigenous and traditional plants, especially fruits, remain essential in the diversification of food in order to enhance food and nutrition security globally [4,42]. Even though the abundance of indigenous plants, especially fruits, is generally well acknowledged in sub-Saharan Africa, the documentation of their uses and associated local knowledge remain inadequate [4,13,23]. This is further worsened by the potential loss of valuable local fruit trees due to diverse threats, particularly habitat loss arising from anthropogenic factors [11,51]. The current findings established the rich, indigenous knowledge of local fruit species among participants in the study area. It was also evident (as all the identified plants are often consumed) that these fruits’ species contribute to the food and nutrition security among the participants. Furthermore, these fruits are readily available and accessible to the participants in the study area. Although the many indigenous fruits are often neglected and not well known beyond their immediate environment, increasing awareness and potential exploration for the development of new food and beverage products have been recognized in recent times [8]. This trend is expected to continue given the inherent values, attributes, and traits associated with many indigenous plants [1,4,23].

## 4. Conclusions

The current findings highlight the diverse indigenous knowledge and continuous importance of fruit species among the participants in the study area. Relative to previous work done in Bushbuckridge, four plants (*Canthium inerme*, *Macrotyloma maranguense*, *Searsia penduline,* and *Syzygium intermedium*) are reported in the study area for the first time, to the best of our knowledge. An estimated 48% of the documented fruit species (e.g., *Annona senegalensis*, *Canthium inerme,* and *Sclerocarya birrea*) were well known (FC = 100%) by all the participants. The reported uses for the plants were categorized as food (59%), medicine (34%), material (4%), fuel (1%), social uses (1%), and vertebrate poison (1%). Particularly, *Brechemia discolor*, *Parinari capensis*, *Parinari curatellifolia,* and *Sclerocarya birrea* were considered as the most valuable fruit species based on their high UR. *Strychnos madagascariensis*, *Bridelia micrantha,* and *Berchemia discolor* had the highest (0.4–0.56) UV and regarded as fruit species with the most diverse uses among the participants. The majority of the fruit species are available during summer and mainly sourced from the wild. Overall, these fruit species have the potential to add value in the food production industry, economy, and well-being of people in the local communities. Thus, they can help in addressing some of the United Nations Sustainable Development Goals (UN SDGs) such as Nos. 1, 2, and 3, which focus on no poverty, zero hunger, and good health and well-being, respectively.

## Figures and Tables

**Figure 1 foods-09-01581-f001:**
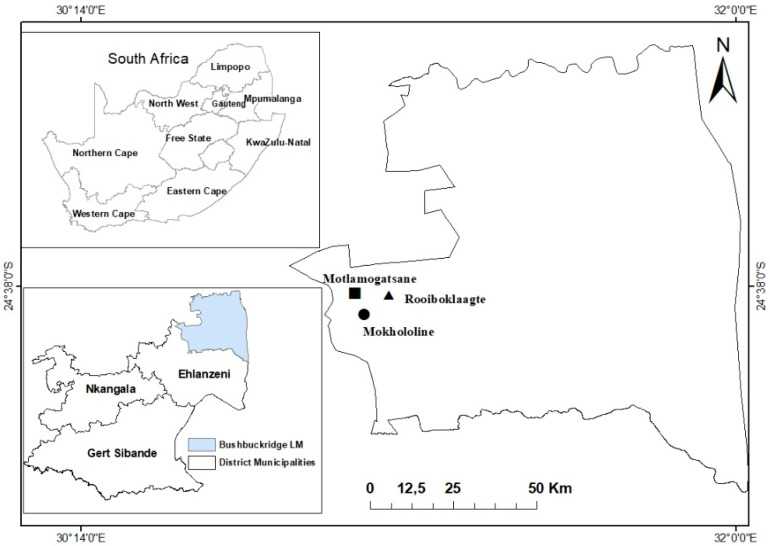
Location of Mokhololine, Motlamogatsane, and Rooiboklaagte B in Bushbuckridge Local Municipality (LM), Ehlanzeni District, Mpumalanga Province, South Africa.

**Figure 2 foods-09-01581-f002:**
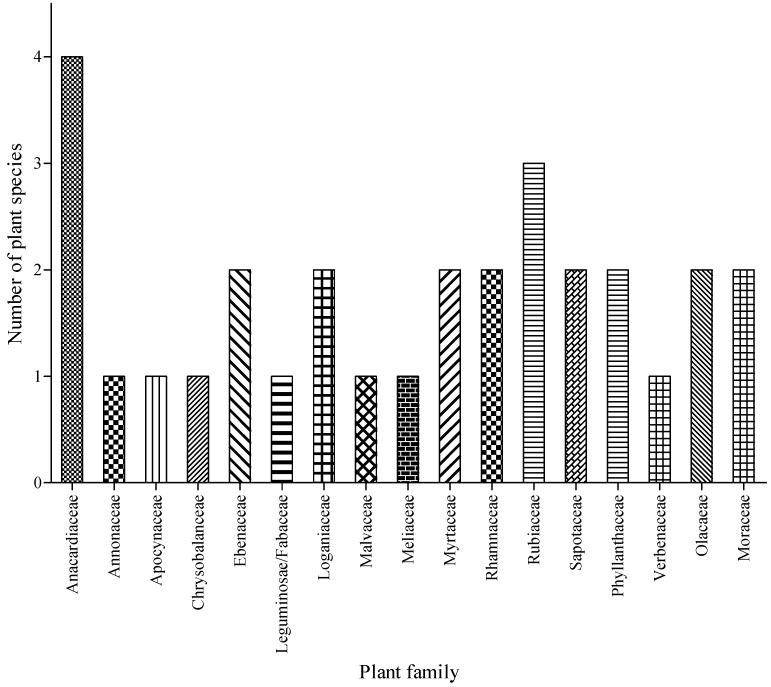
Families of fruit species identified in three villages of Bushbuckridge Local Municipality, Mpumalanga province, South Africa.

**Figure 3 foods-09-01581-f003:**
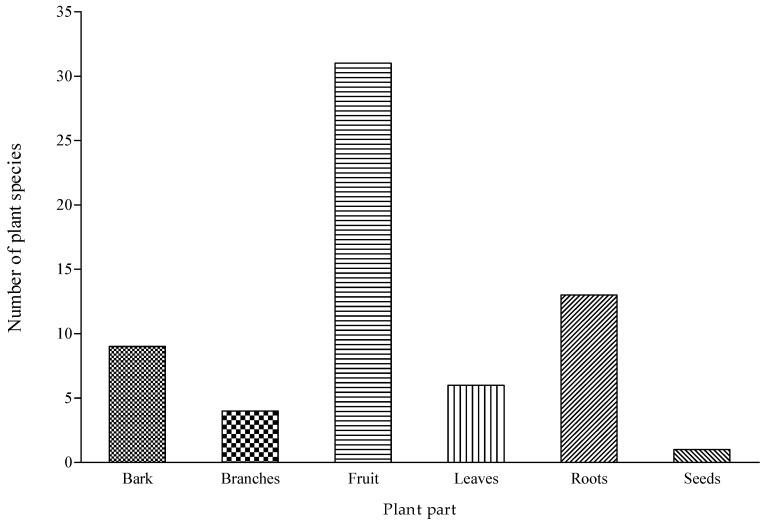
Plant parts of fruit trees used for different purposes in the three villages of Bushbuckridge Local Municipality, Mpumalanga province, South Africa.

**Figure 4 foods-09-01581-f004:**
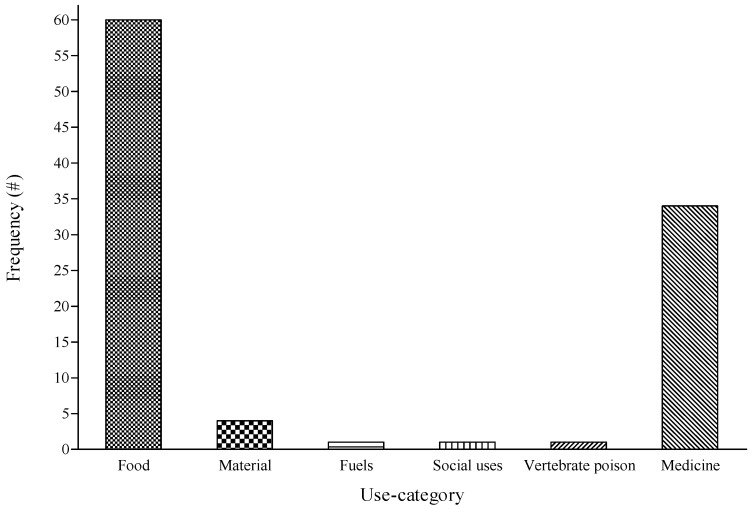
Distribution of use category of fruit species identified in three local communities of Bushbuckridge Local Municipality, Mpumalanga province, South Africa. # denotes number of mentions. Use category was based on the guideline prepared by Cook [28].

**Table 1 foods-09-01581-t001:** Demographic characteristics of the participants in three (3) villages of Bushbuckridge Local Municipality, Mpumalanga province, South Africa.

Feature	Frequency (n)	Percentage (%)
Age groups		
20–35	9	22
36–64	13	32
>64	19	46
Gender		
Male	17	41
Female	24	59
Formal educational level		
None	7	17
Primary	4	10
Secondary	24	58
Tertiary	6	15
Work status		
Employed	5	12.2
Unemployed	5	12.2
Self-employed	10	24.4
Retired	21	51.2

**Table 2 foods-09-01581-t002:** An inventory of locally sourced fruit species among three villages in Bushbuckridge Local Municipality, Mpumalanga Province, South Africa. Use value (UV), frequency of citation (FC), and use report (UR).

Scientific Name and Voucher No	Family	Local Name (Sepulana)	Plant Part(s)	Uses and Significance among Participants	No of Use Category	No of Use Report (UR)	FC (%)	UV
*Annona senegalensis* Pers KNS 10	Annonaceae	Matllepo	Fruit, leaves, roots	Food (edible fruit), vertebrate poison (snake repellent), medicine (cosmetics and sexually transmitted infections/diseases),	3	4	100	0.098
*Berchemia discolor* (Klotzsch) Hemsl KNS 31	Rhamnaceae	Digokgoma	Fruit, bark, roots	Food (edible fruit, porridge and alcoholic beverage), fuel (firewood), medicine (wounds, bleeding gums)	3	6	36.5	0.401
*Berchemia zeyheri* (Sond.) Grubov KNS 20	Rhamnaceae	Dinee	Fruit, roots	Food (edible fruit), medicine (headache)	2	2	24	0.203
*Bridelia micrantha* (Hochst.) Baill. KNS 1	Phyllantaceae	Ditsere	Fruit, bark	Food (edible fruit), medicine (diarrhoea, headache, sore eyes and tooth-ache)	2	5	24	0.508
*Canthium inerme* (L.f.) Kuntze KNS 8	Rubiaceae	Mmitswa	Fruit, branches	Food (edible fruit), material (construction)	2	2	100	0.049
*Carissa edulis* Vahl. KNS 12	Apocynaceae	Dithokolo	Fruit, roots	Food (edible fruit and beverage/wine), social use (misfortunes), medicine (body pains)	3	4	100	0.098
*Diospyros mespiliformis* Hochst. ex A.DC. KNS 5	Ebenaceae	Ditsoma	Fruit, branches	Food (edible fruit), material (wooden spoon), medicine (epilepsy)	3	3	100	0.073
*Englerophytum magalismontanum* (Sond.) T.D. Penn. KNS 9	Sapotaceae	Ditlhatjwa tsa tlhaga	Fruit	Food (edible fruit, jam, syrup and beverage/wine)	1	4	51	0.191
*Euclea divinorum* Hiern KNS 22	Ebenaceae	Motlakolane	Fruit	Food (edible fruit)	1	1	73	0.033
*Ficus sur* Forssk. KNS 3	Moraceae	Mago	Fruit, bark	Food (edible fruit and jam), medicine (chest problems)	2	3	70.3	0.104
*Ficus thonningii* Blume KNS 23	Moraceae	Mokumo (Dikumo)	Fruit	Food (edible fruit and jam)	1	2	46	0.106
*Flueggea virosa* (Roxb. Ex Willd.) Royle subsp. *virosa* KNS 7	Phyllanthaceae	Motlhalabu (Ditlhalabu)	Fruit, bark	Food (edible fruit), medicine (chest problems)	2	2	100	0.049
*Grewia flavescens* Juss KNS 27	Malvaceae	Dipharatshwena	Fruit	Food (edible fruit, beverage/juice)	1	2	51	0.096
*Lannea edulis* (Sond.) Engl. KNS 11	Anacardiaceae	Diphiroku	Fruit, branches	Food (edible fruit), material (wooden spoon)	2	2	100	0.049
*Lannea schweinfurthii* Engl. KNS 29	Anacardiaceae	Marulanopsane	Fruit, branches	Food (edible fruit), material (wooden spoon)	2	2	46	0.106
*Lantana rugosa* Thunb. KNS 34	Verbenaceae	Molutoane	Fruit, leaves, roots	Food (edible fruit), medicine (fever and diarrhoea)	2	3	100	0.073
*Macrotyloma maranguense* (Taub.) Verdc KNS 6	Leguminosae (Fabaceae)	Mokorola kgogo (Mokorolakgogo)	Fruit	Food (edible fruit)	1	1	100	0.024
*Mimusops zeyheri* Sond. KNS 19	Sapotaceae	Mmupudu	Fruit	Food (edible fruit)	1	1	100	0.024
*Parinari curatellifolia* Planch. ex Benth KNS 2	Chrysobalanaceae	Dipola	Fruit, roots	Food (edible fruit, soft porridge and syrup), medicine (pneumonia, sore eyes and ear problems)	2	6	51	0.287
*Parinari capensis* Harv. KNS 30	Chrysobalanaceae	Dimola	Fruit, roots	Food (edible fruit, soft porridge and syrup), medicine (ear problem and sore eyes)	2	6	51	0.287
*Sclerocarya birrea* (A.Rich.) Hochst. subsp. *caffra* (Sond.) KNS 28	Anacardiaceae	Morula	Fruit, seeds, bark	Food (edible fruit, jam, nuts and beverage/wine), medicine (sexually transmitted infections/diseases and diarrhoea)	2	6	100	0.146
*Searsia pendulina* (Jacq.) Moffett KNS 25	Anacardiaceae	Botlhotlho	Fruit	Food (edible fruit, alcoholic beverage)	1	2	36.5	0.134
*Strychnos madagascariensis* Poir. KNS 17	Loganiaceae	Magwagwa	Fruit, leaves, roots	Food (edible fruit and sweets), medicine (poison and wounds)	2	5	21.9	0.557
*Strychnos spinosa* Lam. KNS 21	Loganiaceae	Mashala	Fruit, leaves, roots	Food (edible fruit), medicine (poison)	2	2	73	0.067
*Syzygium cordatum* Hochst. ex Krauss KNS 18	Myrtaceae	Ditlho tsa tlhaga	Fruit, roots	Food (edible fruit, alcoholic beverage), medicine (tuberculosis)	2	3	100	0.073
*Syzygium intermedium* Engl. & Brehmer KNS 4	Myrtaceae	Ditlho	Fruit	Food (edible fruit)	1	1	100	0.024
*Trichilia emetica* Vahl subsp. *emetica* KNS 13	Meliaceae	Mogotlho	Fruit, bark, roots	Food (edible fruit, dried fruit, beverage/juice), medicine (kidney problems)	2	4	48.7	0.200
*Vangueria infausta* Burch. KNS 16	Rubiaceae	Mabilo	Fruit, bark, roots	Food (edible fruit, dried fruit, alcoholic beverage), medicine (sexually transmitted infections/diseases and tooth-ache)	2	5	100	0.122
*Vangueria pygmaea* Schltr. KNS 15	Rubiaceae	Mmilofasane Mabilofasane	Fruit, bark, leaves	Food (edible fruit, dried fruit, alcoholic beverage), medicine (sexually transmitted infections/diseases)	2	4	100	0.098
*Ximenia americana* L. KNS 24	Olacaceae	Ditsadi	Fruit, bark, roots	Food (edible fruit), medicine (wounds and constipation)	2	3	21.9	0.334
*Ximenia caffra* Sond. KNS 14	Olacaceae	Dichidi	Fruit, leaves	Food (edible fruit and jam), medicine (diarrhoea, infertility, fever)	2	5	100	0.122

**Table 3 foods-09-01581-t003:** Seasonal availability (W = winter, Sp = spring, Su = summer, A = autumn), occurrence, and ecological traits of fruit species in three villages in Bushbuckridge Local Municipality, Mpumalanga province, South Africa. SATN = South Africa. ^$^ Life-form number provided in bracket is based on National List of Indigenous Trees (https://www.treetags.co.za/national-list-of-indigenous-trees/). ^#^ Colors of ripe fruit were similar to the description by Mashile et al. [21]

Scientific Name	Availability	Occurrence	^$^ Life-Form	^#^ Colour Of Ripe Fruit	Fruiting Period	Source
*Annona senegalensis*	Su	Common	Shrub (105)	Yellow	Dec–Mar [6]	Wild
*Berchemia discolor*	Su	Common	Tree (449)	Yellow	Jan–May [53]	Wild
*Berchemia zeyheri*	Su, A	Common	Tree (450)	Red	Jan–May [53]	Wild
*Bridelia micrantha*	W	Common	Tree (324)	Black	Oct–Dec [53]	Wild
*Canthium inerme*	Sp, Su	Common	Shrub (708)	Brown	Oct–Apr [54]	Wild
*Carissa edulis*	Sp, Su	Common	Shrub (604.4)	Black	Sept–Dec [55]	Wild
*Diospyros mespiliformis*	Sp, Su	Common	Tree (606)	Brown	Apr–Sep [46]	Wild
*Englerophytum magalismontanum*	W, Su, A	Common	Tree (581)	Red	Dec–Feb [47]	Wild
*Euclea divinorum*	Sp, Su	Common	Shrub (595)	Black	Dec–Mar [54]	Wild
*Ficus sur*	W, Sp, Su, A	Common	Tree (50)	Red	Sept–Mar [54]	Wild
*Ficus thonningii*	W, Sp, Su, A	Common	Tree	Red	Oct–Dec [46]	Wild
*Flueggea virosa*	Sp, Su	Common	Shrub (309)	White	Dec–Mar [54]	Wild
*Grewia flavescens*	Su	Common	Shrub (459.2)	Brown	Dec–Mar [54]	Wild
*Lannea edulis*	Su	Common	Shrub	Black	Oct–Dec [46]	Wild
*Lannea schweinfurthii*	Su	Common	Tree (363)	Yellow	Nov–Mar [54]	Wild
*Lantana rugosa*	Sp, Su	Common	Shrub	Purple	Sep–May [48]	Domesticated
*Macrotyloma maranguense*	Su	Common	Shrub	Yellowish or pinkish-brown	Sep–May [48]	Domesticated
*Mimusops zeyheri*	Su	Common	Tree (581)	Yellow	Apr–Sep [54]	Wild
*Parinari capensis*	W, Sp	Common	Tree	Brown	Oct–Jan [54]	Wild
*Parinari curatellifolia*	Sp	Common	Shrub (146)	Brown	Oct–Nov [6]	Wild
*Sclerocarya birrea*	Su	Common	Tree (360)	Yellow	Feb–Jun [54]	Wild
*Searsia pendulina*	W, A	Common	Tree (396)	Black	Dec–May [54]	Wild
*Strychnos madagascariensis*	Su	Common	Tree (626)	Yellow	Sept–Feb [49]	Wild
*Strychnos spinosa*	Sp, Su	Common	Tree (629)	Yellow	Sept–Feb [49]	Wild
*Syzygium cordatum*	Sp, Su	Common	Tree (551)	Deep purple	Nov–Mar [6]	Wild
*Syzygium intermedium*	A	Common	Tree (556.1)	Red	Dec–Apr [54]	Domesticated
*Trichilia emetica*	Su, A	Common	Tree (301)	Red to brown	Jan–May [54]	Wild
*Vangueria infausta*	Su, A	Common	Tree (702)	Brown	Jan–May [50]	Wild
*Vangueria pygmaea*	Sp, Su	Common	Tree	Black	Feb–Apr [6]	Wild
*Ximenia americana*	Sp, Su	Common	Shrub (101.5)	Orange	Jan–May [54]	Wild
*Ximenia caffra*	Sp, Su	Common	Shrub (103)	Red	Aug–Oct [54]	Wild

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
