# Peer review of "An Exploratory Study on the Diverse Uses and Benefits of Locally-Sourced Fruit Species in Three Villages of Mpumalanga Province, South Africa"

_foods, 2020, doi:10.3390/foods9111581_

Round 1

Reviewer 1 Report

  • Mark the other color of the county understudy in Figure 1 (bottom right card). It is currently challenging to understand where these villages were located on the map.
  • You should also write about how many people at all live in Bushbuckridge Local Municipality and how many villages are there in total?
  • What language natives speak there villages, and in which language were the interviews conducted? Was it English?
  • To my knowledge, all trees are burning. So you should justify why people named you so few taxa of firewood? For example, Berchemia discolor is used, but another tree in the same genus, Berchemia zeyheri, is not used? What is the reason for the low number of wood species used for heating? Do in ths area area all houses have gas cooker or electric cooker?
  • A lot of tree species can also be used in construction. It is doubtful that they are so little used there. The same applies to other uses for trees, why only Lannea sp. and Diospyros mespiliformis can be used fo making wooden spoons?
  • This sentence could still be supplemented by more countries than now only three countries „The importance of fruit trees among local communities have been documented in countries such as Zimbabwe by Maroyi [29], Ethiopia [30] and in India [31, 32].”. With a simple search, I received the following article:

Sõukand, R., & Kalle, R. (2016). Perceiving the biodiversity of food at chest-height: Use of the fleshy fruits of wild trees and shrubs in Saaremaa, Estonia. Human Ecology44(2), 265-272.

  • Table 3 shows when people pick these fruits. It would be advisable to add another table to the text, where more information on these species could be read. It should be based on botanical literature: when these trees and shrubs bloom; how tall do they grow; how widespread they are in the area (common or rare); is it a native species, a cultivated species or a semi-domesticated or naturalized species? Should also discuss whether there is a relationship between the distribution of tree species and their use; or is there a connection between their height and use, (e.g., lower shrubs are used less than taller trees)?
  • Table 2 shows that beer is made from Searsia pendulina and Berchemia discolor fruits. Nowhere in the text is specified whether cereal malt will be included in this beer? If people do not put malt, please specify the technology, how this beer-like drink is made and why it is called beer (the term beer is very definite https://en.wikipedia.org/wiki/Beer )?
  • You need to spell out in the text what people meant by the term: “Sexually transmitted infections / diseases”?
  • It is trivial in ethnobotany that local traditional knowledge is influenced by books on plants, mass media, school education, etc. Why, then, do you claim that knowledge came only from older people: "Knowledge on how to harvest and process edible fruit species is transmitted from the elderly to the younger generation through community teachings"? How did you find out? Table 1 shows that only 7 people had not attended school, so most of the informants could read. Are in this county (country?) missing botanical literature, where would be the description and teaching how to use of trees? If such literature has been published, it should also be mentioned in the article. What are the most important books that have published teachings on the use of tree species?

Author Response

Reviewer 1

Dear Reviewer, we thank you for making time to review our manuscript an the useful suggestions which has surely improved our manuscript.

Comment: Mark the other color of the county understudy in Figure 1 (bottom right card). It is currently challenging to understand where these villages were located on the map.

Response: Thank you for the observation, due to technical difficulty (beyond our control) the map cannot be amended at the moment but we have added the GPS co-ordinates for all the selected villages in Figure 1 and in the main text (see page 3; line 104-106). We hope this amendment is satisfactory.

Comment: You should also write about how many people at all live in Bushbuckridge Local Municipality and how many villages are there in total?

Response: We have included additional information on the number of people and villages found in Bushbuckridge municipality(see page 3; line 90-99)

Comment: What language natives speak there villages, and in which language were the interviews conducted? Was it English?

Response: The native speak Sepulana (Northern Sotho sub-language) and this was used during the interviews (see page 4; line 119-121)

Comment: To my knowledge, all trees are burning. So you should justify why people named you so few taxa of firewood? For example, Berchemia discolor is used, but another tree in the same genus, Berchemia zeyheri, is not used? What is the reason for the low number of wood species used for heating? Do in this area all houses have gas cooker or electric cooker?

Response: Indeed, this was an interesting observation and may be related to the legislation that discourage the indiscriminate harvesting of tree in many Provinces in South Africa. As an example of this legistalation, kindly see (https://www.gov.za/about-sa/forestry)

The National Forests Act, 1998 (Act 84 of 1998), and the Forestry Laws Amendment Act, 2005 (Act 35 of 2005), reflect the vision for the future of forestry in South Africa.

Secondly, permits are required for removal of indigenous floras, please see (https://www.environment.gov.za/projectsprogrammes/bioprospectingaccess_benefitsharing_babs_clearinghouse)

The National Environmental Management: Biodiversity Act, 10 of 2004 (NEMBA)

Comment: A lot of tree species can also be used in construction. It is doubtful that they are so little used there. The same applies to other uses for trees, why only Lannea sp. and Diospyros mespiliformis can be used for making wooden spoons?

Response: As highlighted in the aforementioned response, additional probing questions on other uses besides the food benefits were not extensively probed.

Comment: This sentence could still be supplemented by more countries than now only three countries „The importance of fruit trees among local communities have been documented in countries such as Zimbabwe by Maroyi [29], Ethiopia [30] and in India [31, 32].” With a simple search, I received the following article:

Sõukand, R., & Kalle, R. (2016). Perceiving the biodiversity of food at chest-height: Use of the fleshy fruits of wild trees and shrubs in Saaremaa, Estonia. Human Ecology,44, 265-272.

Response: Thanks for the input, other counties such as Estonia and Indonesia were included to enhance the sentence (see page 6; line 177-179). This new addition also reflect on the widespread trend across different continents.

Comment: Table 3 shows when people pick these fruits. It would be advisable to add another table to the text, where more information on these species could be read. It should be based on botanical literature: when these trees and shrubs bloom; how tall do they grow; how widespread they are in the area (common or rare); is it a native species, a cultivated species or a semi-domesticated or naturalized species? Should also discuss whether there is a relationship between the distribution of tree species and their use; or is there a connection between their height and use, (e.g., lower shrubs are used less than taller trees)?

Response: We sincerely appreciate the sugestions provided but strongly feel that the inclusion of some of the aspects will be outside the scope of the current study. Nevertheless, we have added additional information in the revised Table 3 that contains some of the information suggested. Some sentences have also been added to the discussion to highlights the new addition.  

Comment: Table 2 shows that beer is made from Searsia pendulina and Berchemia discolor fruits. Nowhere in the text is specified whether cereal malt will be included in this beer? If people do not put malt, please specify the technology, how this beer-like drink is made and why it is called beer (the term beer is very definite https://en.wikipedia.org/wiki/Beer)?

Response: Unfornuately, we did not probe on the technology used for preparing the different beverages. The scope of the current research did not include technology or methods of preparing the documented fruit species.

Comment: You need to spell out in the text what people meant by the term: “Sexually transmitted infections/diseases”?

Response: We have given examples of the “Sexually transmitted infections/diseases” in the revised manuscript (see page 15; line 261-262)

Comment: It is trivial in ethnobotany that local traditional knowledge is influenced by books on plants, mass media, school education, etc. Why, then, do you claim that knowledge came only from older people: "Knowledge on how to harvest and process edible fruit species is transmitted from the elderly to the younger generation through community teachings"? How did you find out? Table 1 shows that only 7 people had not attended school, so most of the informants could read. Are in this county (country?) missing botanical literature, where would be the description and teaching how to use of trees? If such literature has been published, it should also be mentioned in the article. What are the most important books that have published teachings on the use of tree species?

Response: Due to the lack of clarity and limited evidence, we have deleted the highlighted sentence.

Reviewer 2 Report

Nice study. However, as it is limited to the fruiting trees only, it would be necessary to put it in the ecological context and the availability of the trees in the region. I would suggest you have a look at the attached article (esp. Table 2) in order to gain ideas on how you could comparatively present your data. You could also add the nutritive data, where available. 

"The utilisation of indigenous plants is relatively low mainly due to lack of knowledge on their nutrient value resulting from limited research"  - please do  not repeat nonsense taken from other sources without critical filter - since what time indigenous use of plants has been influenced by research? In this case we will have local ecological knowledge (not indigenous), derived from books, but this is certainly not the main source of knowledge (although books may have some influence). 

Figure 3: much more interesting would be to see the proportion of URs, not just plant parts. Calculate the number of URs (use records) also for the uses as well.

In Table 2, please indicate the frequency of citation for every specific use, not only the conglomerate of all uses. 

For Table 3 you'd need to indicate also the fruiting time of the trees indicated in ecological and botanical literature. If there are any discrepancies, this would be a very nice piece for discussion

Figures 2 and 3: why do you use different patterns? No need for that. 

Figure 2: what do you mean by Frequency? The number of species in the Family? The table makes sense only if it will show a comparison between the numbers of used and not used taxa in the family.

As a result, you should build your conclusion on the availability of the resource and actual use and not on the description of obtained data (which is not really that spectacular and does not answer any serious scientific question. The scientific research question needs to be stated at the end of the introduction; the description alone is not enough.  

The paper has great potential, but now is slightly below average. 

Author Response

Reviewer 2

Nice study. However, as it is limited to the fruiting trees only, it would be necessary to put it in the ecological context and the availability of the trees in the region. I would suggest you have a look at the attached article (esp. Table 2) in order to gain ideas on how you could comparatively present your data. You could also add the nutritive data, where available. 

RESPONSE: Thank you for the suggested article, it has provided us with valuable information that was used to enhance the presentation and discussion of our manuscript. 

"The utilisation of indigenous plants is relatively low mainly due to lack of knowledge on their nutrient value resulting from limited research"  - please do  not repeat nonsense taken from other sources without critical filter - since what time indigenous use of plants has been influenced by research? In this case we will have local ecological knowledge (not indigenous), derived from books, but this is certainly not the main source of knowledge (although books may have some influence). 

RESPONSE: Thanks for the observation, we have deleted the sentence.

Figure 3: much more interesting would be to see the proportion of URs, not just plant parts. Calculate the number of URs (use records) also for the uses as well.

RESPONSE: Figure 3 is meant to indicate the overall distribution of the 31 fruit species in the study area. Specific uses linked to the different plant part is indicated in Table 2

In Table 2, please indicate the frequency of citation for every specific use, not only the conglomerate of all uses. 

RESPONSE: As recommended, we have included the frequency for every specific uses (this include the number of category and the individual use reports, URs), please see the revised Table 2

For Table 3 you'd need to indicate also the fruiting time of the trees indicated in ecological and botanical literature. If there are any discrepancies, this would be a very nice piece for discussion

RESPONSE: As recommended, we have included the fruiting time for the fruit trees based on existing ecological and botanical literature. Additional sentence focusing on this aspect has been included in the revised manuscript. Please see section 3.4. 

Figures 2 and 3: why do you use different patterns? No need for that. 

RESPONSE: All the Figures have been revised to indicate similar pattern

Figure 2: what do you mean by Frequency? The number of species in the Family? The table makes sense only if it will show a comparison between the numbers of used and not used taxa in the family.

RESPONSE: We are of the opinion that the Figure gives a summary of the distribution of the families for the fruit species used in the study area. We highlighted the similarities and differences relative to existing literature.  

As a result, you should build your conclusion on the availability of the resource and actual use and not on the description of obtained data (which is not really that spectacular and does not answer any serious scientific question. The scientific research question needs to be stated at the end of the introduction; the description alone is not enough.  

RESPONSE: We have revised the conclusion and clearly indicated the availability of resources and the actual use. Research questions are also highlighted at the end of the introduction as suggested

The paper has great potential, but now is slightly below average. 

RESPONSE: Thank you for the opportunity to revise our manuscript. We are grateful for the valuable suggestions which has enhanced the quality of our manuscript. We trust you will find our comments/responses satisfactory.

Reviewer 3 Report

Dear Authors,

I find the article very interesting especially in relation to sustainable development goals. There are just a few small suggestions that you can find in the manuscript (pdf version).
After these minor revisions the paper is, in my opinion, suitable for the publication.

Best Regards

Author Response

Comment: In the conclusions are mentioned the United Nations Sustainable Development Goals (UN SDGs), would be interesting and more comprehensive if this topic were also briefly covered in the introduction.

Response: The United Nations Sustainable Development Goals (UN SDGs), was briefly covered in the introduction (see page 2; line 72-73).

Comment: Please change with: “helping to solve"

Response: The word helping was revised to “solve” (see page 2; line 49)

Comment: Please, write this sentence better

Response: The sentence in page 1; line 50-51 was revised to enhance the clarity to readers.

Comment: Please, correct the degree symbol

Response: the correct degree symbol has been revised (see page 3; line 100)

Comment: It is not correct to quote here the table

Response: We have removed the reference to the Table based on the recommendation from the reviewer

Round 2

Reviewer 1 Report

I thank you for your in-depth explanation of the local situation!

One technical note, you use one term "indigenous" to describe both native people and native plants in one sentence. This makes the sentence construction cumbersome, e.g., "Since time immemorial, indigenous communities have relied on diverse indigenous plant species because they are easily accessed from their immediate environment."; "However, the indigenous knowledge on indigenous fruit species is diminishing due to urban migration as well as preference for exotic fruits. ". Couldn't some synonym (local, native) be used in these sentences: "Since time immemorial, indigenous communities have relied on diverse native plant species because they are easily accessed from their immediate environment." and so on.

Author Response

I thank you for your in-depth explanation of the local situation!

One technical note, you use one term "indigenous" to describe both native people and native plants in one sentence. This makes the sentence construction cumbersome, e.g., "Since time immemorial, indigenous communities have relied on diverse indigenous plant species because they are easily accessed from their immediate environment."; "However, the indigenous knowledge on indigenous fruit species is diminishing due to urban migration as well as preference for exotic fruits. ". Couldn't some synonym (local, native) be used in these sentences: "Since time immemorial, indigenous communities have relied on diverse native plant species because they are easily accessed from their immediate environment." and so on.

RESPONSE: Thank you for the comment, we have rephrased the sentences and used synonym to eliminate the cumbersome nature currently evident in the manuscript. See for examples, line 49, 69 and 72

Reviewer 2 Report

Thank you for accepting the majority of recommendations. Just a few notes: 

Please indicate the regions you studied on the general map as dots or draw the county you worked in. Also, the position of your studied region is missing from the map - you cannot assume that reader will search the coordinates. 

Please note that in scientific literature "branches" is not used term, instead, please use "twigs". 

Author Response

Thank you for accepting the majority of recommendations. Just a few notes: 

RESPONSE: Thank you for the valuable recommendation, we sincerely appreciate the inputs.

Please indicate the regions you studied on the general map as dots or draw the county you worked in. Also, the position of your studied region is missing from the map - you cannot assume that reader will search the coordinates. 

 RESPONSE: We have updated the map with clear indication of the villages selected for the current study.

Please note that in scientific literature "branches" is not used term, instead, please use "twigs". 

RESPONSE: We decided to use the term ‘branches’ because of our understanding that ‘twigs’ is different from ‘branches (see notes below). Based on the response from our participants, the branches (and not the twigs) are used for constructions purposes and productions of utensils particularly wooden spoon. To the best of our knowledge, the term ‘branches’ is correct in the current context.